# Rapid Removal of Cr(VI) from Wastewater by Surface Ionized Iron-Based MOF: Ion Branching and Domain-Limiting Effects

Chen Wang, Jiakun Chen and Qi Yang *

School of Water Resources and Environment, China University of Geosciences, Beijing 100083, China; 17603363661@163.com (C.W.); cjk2245986367@outlook.com (J.C.)

* Correspondence: yq@cugb.edu.cn; Tel.: +86-13691491158; Fax: +86-10-82321081

**Abstract:** Exploring the ratio of metal centers to organic ligands and the amount of DMF are important to improve the stability and adsorption efficiency of MOF materials as adsorbents. In this work, MIL101(Fe)-$Na_2CO_3$ was successfully obtained by modification with formic acid, sodium carbonate, carbon nanotubes, and moieties. The adsorption efficiency of MIL-101(Fe) on Cr(VI) was greatly improved, and the removal efficiency was able to reach 100% in 20 min with a maximum adsorption capacity of 20 mg/g. The inhibition order of the competing anions for the removal of hexavalent chromium was as follows: $Cl^- < NO_3^- < SO_4^{2-}$. The analysis of the adsorption thermodynamic model found that the adsorption of MIL101(Fe)-$Na_2CO_3$ for Cr(VI) showed spontaneous heat-absorbing and entropy-increasing chemisorption behavior. When using NaOH as the eluent and HCl as the regeneration stabilizer, MIL-101(Fe)-$Na_2CO_3$ had good adsorption capacity in multiple cycles.

**Keywords:** MOF materials; Cr(VI); adsorption kinetics; adsorption thermodynamic





## 1. Introduction

Cr(VI) is highly migratory and easily soluble in water, and is considered a carcinogenic and mutagenic agent that is harmful to the ecological environment [1,2]. The toxic effects of Cr(VI) on the human body are mainly in terms of skin irritation, induction of lung cancer, and damage to the kidney, liver, stomach, etc.; the negative effects on the environment are mainly in terms of making certain plants sprout and grow less, causing algae to die, etc. [3–5]. The U.S. Environmental Protection Agency (USEPA) has identified Cr(VI) as one of the most toxic pollutants in the water system, and has published a maximum limit of 0.05 mg/L for Cr(VI) in water [6].

With the deep development of industry, the pollution of Cr(VI) has spread all over the regions and even further affected the life of local people. The diffusivity and toxicological relevance of Cr(VI) itself make water bodies produce a serious and permanent pollution state. Therefore, it is urgent to solve Cr(VI) pollution, and it is important to find a cost-effective Cr(VI) pollution removal technology. Currently, researchers at home and abroad have proposed a variety of methods to remove Cr(VI) from wastewater, mainly including chemical reduction precipitation, adsorption, ion exchange resin method, solvent extraction, membrane separation, etc. [7–11].

Metal-organic frameworks (MOFs) are coordination polymers making up a class of highly ordered porous crystalline hybrid materials consisting of metal clusters and polyfunctional organic linkers. Compared with conventional adsorbents, MOFs present captivating merits due to their varying compositions and structures, such as higher surface area and pore volume, massive porosity, adjustable pore size, and favorable dispersion of metal ions in the framework [12]. Their characteristics include adjustable topology, high porosity, low density, and high thermal and chemical stability [13]. In recent years, MOFs have been used in a wide range of applications such as gas separation/storage, water purification, chemical sensors, optics, drug transport, bioreactors, and adsorption [14–21].

MOFs are effective adsorbents for the removal of contaminants from aqueous solutions because of their extraordinary structural and surface properties [22]. For the adsorption of organic pollutants, various interactions between the host and the guest occur, such as electrostatic, acid-base, H-bonding, hydrophobicity, and coordination with open metal sites, all of which have important effects on adsorption [23,24]. In a previous study, a variety of MOFs were used to adsorb two drugs, furosemide and salazosulfapyridine, in the aqueous phase, then the adsorption efficiency and stability were compared. It was found that MIL-101(Cr) was the most stable and had the highest adsorption capacity, and there was a synergistic effect between the drug and MIL-101(Cr) [25]. Regarding the removal of inorganic pollutants, different MOFs have different adsorption capacities for different metal ions, and numerous studies have been previously performed. AMOF-1 material has been synthesized as an effective adsorbent for Cd(II) with a maximum adsorption capacity of 41 mg/g, while $Fe_3O_4$@MIL-100(Fe) material has been synthesized for Cr(VI) removal with a maximum adsorption capacity of 18 mg/g [26,27]. Therefore, MOFs may be a promising material for adsorption of hexavalent chromium. However, few studies have reported the removal of Cr(VI) using MOFs.

In this paper, we successfully modified MIL-101(Fe)-$Na_2CO_3$ by controlling and deploying the ratio of metal centers to organic ligands and the DMF dosage of MIL-101 in order to realize the rapid degradation of Cr(VI) through ionic branching and domain-limiting effects. In addition, this paper clarifies the deep-seated mechanism (electrostatic attraction, adsorption on monomolecular layer) for the removal of Cr(VI), which reveals the potential role of modified MIL-101 in practical applications.

MIL-101(Fe) MOFs were selected as the main object of study due to the pollution status of Cr(VI), and were applied to the adsorption of Cr(VI) in water through appropriate modification in order to achieve a better adsorption effect. First, we sought to investigate the effects of the ratio of metal center to organic ligand, DMF dosage, and reaction time on the stability and adsorption efficiency of the synthesized adsorbent and to select the best conditions for the synthesis of MIL-101(Fe). Second, MIL-101(Fe)-$Na_2CO_3$ adsorbent material was successfully obtained by modification with formic acid, sodium carbonate, carbon nanotubes, and groups, and its morphology was characterized. Third, the adsorption efficiency of the adsorbent for Cr(VI) was investigated by kinetic and thermodynamic adsorption models.

## 2. Materials and Preparation

### 2.1. Chemicals

Sodium hydroxide (NaOH, $\geq$98.0%), hydrochloric acid (HCl, $\geq$98.0%), anhydrous ethanol (EtOH), potassium dichromate (Potassium dichromate standard solution, 0.0999 mol/L), acetone ($C_3H_6O$, $\geq$99.5%), diphenylcarbonyl dihydrazide ($C_{13}H_{14}N_4O$, AR), sulfuric acid ($H_2SO_4$, 95.0~98.0%), phosphoric acid ($H_3PO_4$, $\geq$85.0%), ferric chloride ($FeCl_3$, 98.0%), terephthalic acid ($H_2BDC$, $\geq$99.0%), amino-terephthalic acid ($C_8H_7NO_4$, $\geq$98.0%), formic acid (GCS, $\geq$99.5%), N,N-dimethylformamide (DMF, 99.8%), acetic acid (Electronic grade G2), and zinc nitrate hexahydrate (Sigma-228737, 98%) were from Sinopharm (Shanghai, China).

### 2.2. Adsorbent Preparation

The materials were prepared using a hydrothermal autoclave; the preparation flowchart is shown in Figure 1, and the preparation method was as follows.

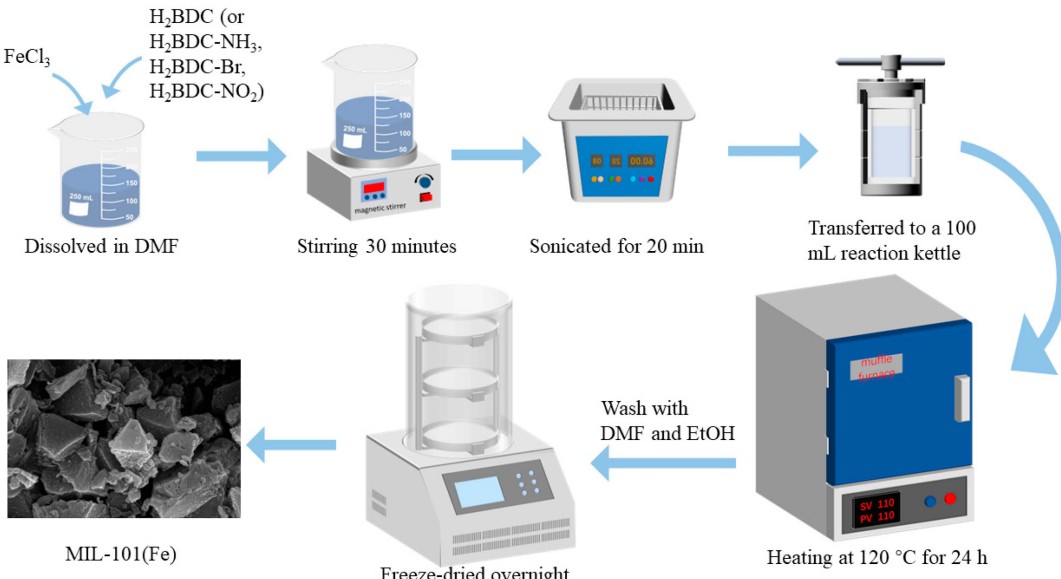

**Figure 1.** Hydrothermal autoclave preparation process.

### 2.2.1. MIL-101 Preparation

A certain amount of anhydrous ferric chloride (4.9 mM) was added to 15 mL of N,N-dimethylformamide (DMF) solvent and a certain amount of terephthalic acid (2.45 mM) was added to 15 mL of DMF; the above two mixed solutions were sonicated for 20 min until completely dissolved. Then, the two solutions were mixed and sonicated for 20 min until they were completely mixed, after which the mixed solutions were transferred to a 100 mL reaction kettle and the temperature of the kettle was maintained at 120 °C for 24 h. After the reaction, the solutions were removed and cooled to room temperature in a ventilated area, a certain speed was set for centrifugal separation, then they were washed three times with hot ethanol and DMF, placed in a vacuum dryer, and freeze-dried to produce MIL-101(Fe) [28].

### 2.2.2. Group Modification

The adsorbent introduction group was prepared mainly by adding a certain amount of amino (nitro, bromine atom)-terephthalic acid (2.45 mM) to 15 mL of DMF solvent and anhydrous ferric chloride (5 mM) to 15 mL of DMF, sonicating the above two solutions for 20 min, then mixing the sonication for 20 min until completely mixed. The mixed solution was then transferred to a 100 mL reactor, the temperature of the reactor was maintained at 120 °C for 24 h, and the reaction was taken out and cooled to room temperature in a ventilated place, centrifuged at a certain speed, washed three times with DMF and hot ethanol, and placed in a vacuum dryer to freeze and dry. This resulted in three MIL-101(Fe) moieties [29].

### 2.2.3. CNT@MIL-101 Preparation

The loaded CNTs were prepared by dispersing different mass fractions of carbon nanotubes (CNTs) into 9 mL of ethanol solvent and sonicating for 10 min until the CNTs were completely dispersed, followed by adding the above-mentioned CNTs with different mass fractions into a DMF mixture of terephthalic acid and ferric chloride and sonicating for 5 min until they were completely mixed. The above mixed solution was transferred to a reactor and maintained at 120 °C for 24 h. After standing and cooling, the solution was centrifuged, washed three times with hot ethanol and DMF, and placed in a vacuum dryer to freeze and dry, i.e., CNT@MIL-101(Fe) loaded with different mass fractions was produced. CNT synthetic adsorbents of 2, 5, 10, 15, and 20 wt% were taken and named as follows:

2CNT@MIL-101(Fe), 5CNT@MIL-101(Fe), 10CNT@MIL-101(Fe), 15CNT@MIL-101(Fe), and 20CNT@MIL-101(Fe).

*2.3. Reaction Conditions and Analytical Methods*

The quantitative Cr(VI) stock solution was placed in the reaction flask. Experiments were carried out by controlling single variables such as the pH, reaction temperature, and initial contaminant concentration The reaction solutions were filtered and sieved for different time periods using a 0.22 μm filter membrane; the pH was adjusted by 0.1 mol/L HCl and 0.1 mol/L NaOH by controlling the thermostatic oscillator temperature to maintain different reaction temperatures. The speed of the thermostatic oscillator was set to 180 rpm, then the concentration of Cr(VI) in the solution was detected by the color development reaction and the removal rate of Cr(VI) and the adsorption amount of the adsorbent were calculated by Equations (1) and (2):

$$\text{Removal efficiency}(\%) \ = \frac{C_0 - C_e}{C_0} \times 100\% \tag{1}$$

$$q_e = \frac{C_0 V_0 - C_e V_e}{m} \tag{2}$$

where $q_e$(mg/g) is the equilibrium adsorption capacity, $C_0$ and $Ce$ (mg/L) are the initial concentration and final concentration of Cr(VI), respectively, $m$ (g) is the mass of dry hydrogel, and $V_0$ and $V_e$ (L) are the initial volume and equilibrium volume during the adsorption process, respectively [30].

In this paper, experiments were conducted by controlling a single variable and all experiments were conducted three times.

## 3. Results and Discussion

*3.1. Characterization*

Figure 2 presents the SEM image of the adsorbent. Figure 2a shows MIL-101(Fe); it can be seen that it has a typical octahedral structure, which is in agreement with previous studies that successfully synthesized MIL-101(Fe) [31]. Figure 2b–d show three different groups; it can be seen that the MIL-101(Fe)-NH2 and MIL-101(Fe)-NO2 particles have good crystallinity that is similar to MIL-101(Fe) and have similar octahedral structures, while MIL-101(Fe)-Br generates a crystal-like layer of MIL-101(Fe) has poorer crystallinity. After the incorporation of carbon nanotubes into the MIL-101(Fe), the morphology of CNT@MIL-101(Fe) shows hybrid characteristics, with CNT entangled in the MIL-101(Fe) crystals (Figure 2e; due to the large size, it could not be clearly photographed at 200 nm; thus, 1 μm was used instead). In addition, the size of CNT@MIL-101(Fe) is increased compared to MIL-101(Fe) due to the attachment and reinforcement of CNT [32].

As can be seen in Figure 3a, MIL-101(Fe) has a better crystalline phase structure and the main characteristic diffraction peaks appear at 2θ of 9.44°, 12.61°, 16.62°, and 18.86°, which is similar to previous results from the literature [33]. The positions of the characteristic diffraction peaks of the MOFs modified by -NH$_2$, -NO$_2$, and carbon nanotubes are almost the same as the positions of the main peaks of MIL-101(Fe), indicating that the addition of CNTs, etc., did not prevent the generation of MIL-101 crystals. A few diffraction peaks in MIL-101(Fe)-Br disappeared, and there were only weak characteristic diffraction peaks at 9.58° and 16.76°, which suggests that the addition of bromine atoms caused a change in the crystal structure of the MIL-101(Fe) system.

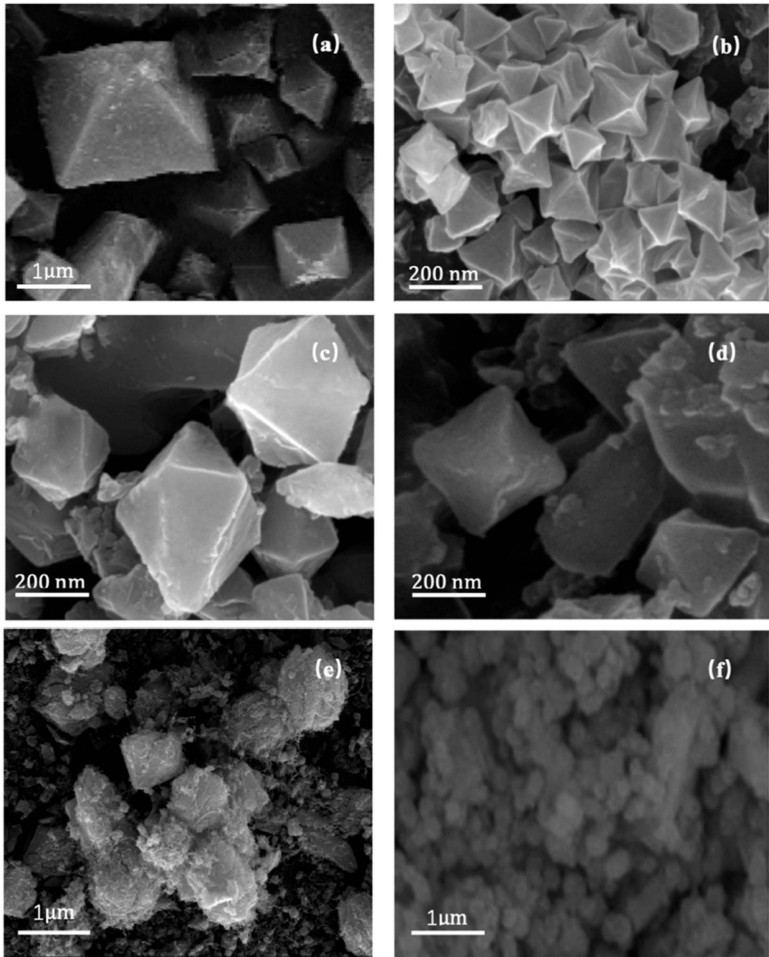

**Figure 2.** SEM images of adsorbents: (**a**) MIL-101(Fe), (**b**) MIL-101(Fe)-NH$_2$, (**c**) MIL-101(Fe)-NO$_2$, (**d**) MIL-101(Fe)-Br, (**e**) 5CNT@MIL-101(Fe), (**f**) MIL-101-Na$_2$CO$_3$.

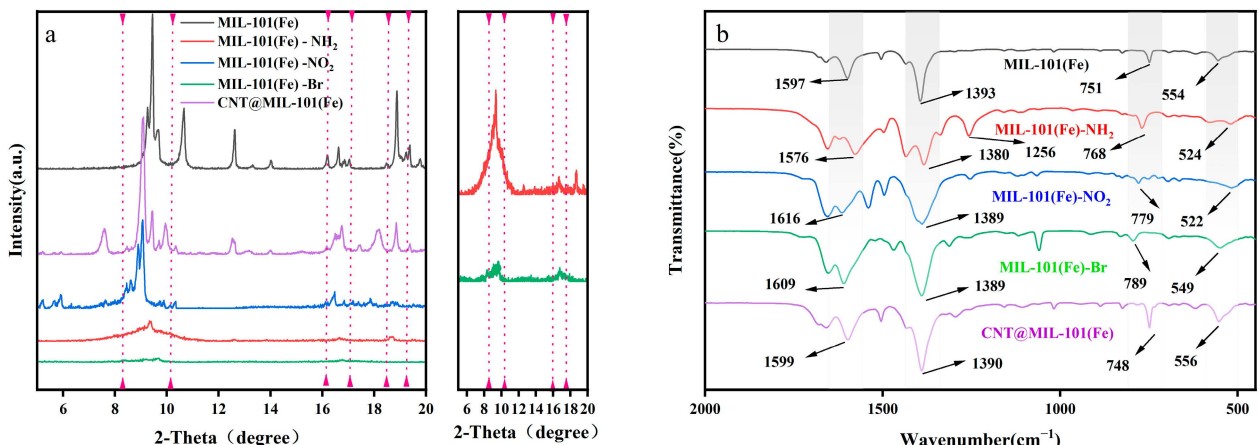

**Figure 3.** (**a**) XRD diffractogram of MOFs (the red dotted line is the MIL-101(Fe) eigenvalue) and (**b**) FT-IR of MOFs.

The FT-IR plots of MIL-101(Fe), MIL-101(Fe)-NH2, MIL-101(Fe)-NO2, MIL-101(Fe)-Br, and 5CNT@MIL-101(Fe) are shown in Figure 3b. MIL-101(Fe) shows a distinct FT-IR pattern, with peaks at 554, 751, 1393, and 1597 cm$^{-1}$. The peaks at 751 cm$^{-1}$ and 554 cm$^{-1}$ indicate the C-H bond [34] and Fe-O bond [35] on the benzene ring, respectively. The bands at 1597 and 1393 cm$^{-1}$ represent the asymmetric and symmetric stretching of O–C–O [36].

These characteristic peaks are present in the remaining four materials as well. In addition to the typical peaks of MIL-101(Fe), the FT-IR spectra may vary due to changes in the ligand functional groups, where in MIL-101(Fe)-NH2 there is an MIL-101(Fe)-deficient peak at the 1256 cm$^{-1}$ band which represents the stretching of aromatic C-N. Upon addition of CNT, no significant shift of the characteristic peak was observed, similar to the characteristic peak of MIL-101(Fe) [37].

Figure 4 shows that the $N_2$ adsorption–desorption isotherms of MIL-101(Fe), MIL-101(Fe)-$NH_2$, MIL-101(Fe)-$NO_2$, MIL-101(Fe)-Br, and 5CNT@MIL-101(Fe) are type I isotherms with $H_4$ hysteresis loops. Five isotherms indicate the presence of meso- and micropores in these samples. Table S1 lists the BET surface areas of the MIL-101(Fe), MIL-101(Fe)-$NH_2$, MIL-101(Fe)-$NO_2$, MIL-101(Fe)-Br, and 5CNT@MIL-101(Fe) samples, which were 1730.85, 1724.85, 1652.52, 2318.37, and 2443.35 $m^2g^{-1}$, respectively. The total pore volumes of the samples were 0.601, 0.511, 1.021, 1.578, and 1.127 $cm^3g^{-1}$, respectively.

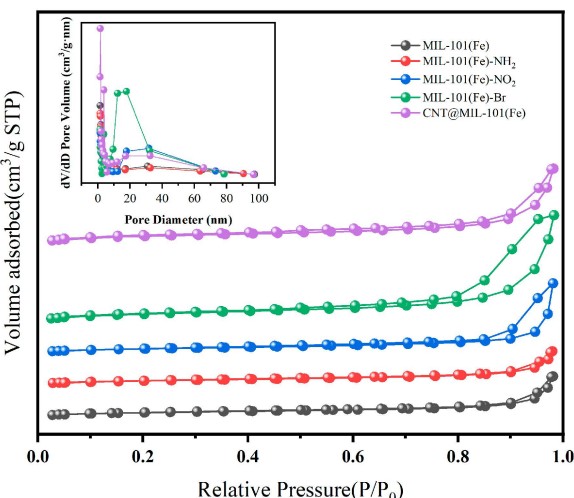

**Figure 4.** $N_2$ adsorption–desorption isotherms of MOFs.

### 3.2. MIL-101 Preparation, Modulation, and Modification

As can be seen from Figure 5a and Table S2, the adsorption effect of the synthesized adsorbent on Cr(VI) varied with the variation of the $n(H_2BDC):n(FeCl_3)$ ratio; the adsorption removal rate of Cr(VI) was the largest when the ratio of $n(H_2BDC):n(FeCl_3)$ was 1:2, reaching 50%. MOFs of the MIL series are structures composed of metal nodes (clusters) and organic connections; thus, the amounts of different ratios of metal to organic ligand are important for the formation of the MOF structure. The reason for the maximum removal rate of Cr(VI) adsorption when the ratio of n(H2BDC):n(FeCl$_3$) is 1:2 may be because the crystallinity of the MIL-101(Fe) adsorbent under this ratio is the highest; with the continuous increase of FeCl$_3$, the excess FeCl$_3$ cannot be fully ligated with the H2BDC connection, resulting in the synthetic MIL-101(Fe) adsorbent. The crystallinity of the synthetic MIL-101(Fe) adsorbent decreases, the purity of the adsorbent decreases, and the adsorption effect then decreases.

According to the structure of MIL-101(Fe) and the chemical stoichiometry characteristics of Fe$_3$OCl-(DMF)$_2$(H2BDC)$_3$, it is known that the amount of DMF has an important influence in the synthesis of MIL-101(Fe) [27]. As described above, the best input ratio of $n(H_2BDC):n(FeCl_3)$ is 1:2; thus, while keeping this ratio constant during the synthesis of MIL-101(Fe), 10 mL, 20 mL, 30 mL, 40 mL, and 50 mL of DMF were added while selecting the same Cr(VI) concentration of 10 mg/L in the adsorption reaction. Under the condition of ensuring the minimum dissolved amount, the amount of DMF was increased and the adsorption effect was observed. It can be seen from Figure 5b and Table S3 that the adsorption and removal rate of Cr(VI) changed significantly with the increasing amount of added DMF; the adsorption rate increased by about 10% when the amount of DMF added was

20 mL, and reached 71% when the amount of DMF added was 30 mL, while the adsorption rate was maintained at about 72% when the amount of DMF added was 40 mL and 50 mL.

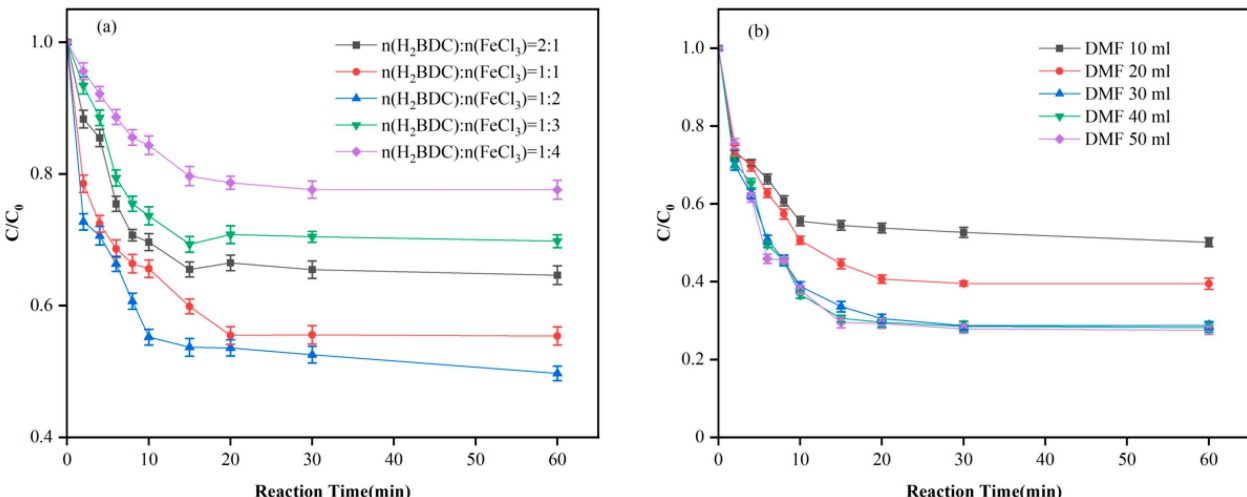

**Figure 5.** (**a**) Effect of the ratio of metal to organic ligand synthesis on the removal rate of Cr(VI); (**b**) effect of different DMF dosages on the removal rate of Cr(VI) (pH = 5.77, T = 30 °C, dosage of adsorbent = 1 g/L, concentration of Cr(VI) = 10 mg/L).

When 10 mL and 20 mL of DMF were added, it was difficult to dissolve the organic ligand completely, affecting the coordination of the organic ligand and metal and failing to achieve sufficient connection coordination, thereby affecting the crystallinity of the adsorbent. With the addition of 30 mL of DMF, the organic ligand and metal center were fully dissolved and coordinated, the best ratio of $n(H_2BDC):n(FeCl_3)$ could be maintained, and the best adsorption effect was obtained. It is worth noting that excess DMF dilutes the concentration of both the metal and the organic ligand, which has a certain hindering effect on their coordination. When the dosage of DMF continues increasing to excess, this further dilutes the concentration of metals and organic ligands; the resulting hindering effect on the coordination of metals and organic ligands means that the adsorption effect grows insignificantly.

Introducing functional groups on the adsorbent may increase certain active adsorption sites and strengthen the bonding energy between the adsorption bonds. However, it was found in our adsorption experiments (Figure 6a) that the adsorption and removal rate of pollutants decreased relative to the initial MIL-101(Fe) after the introduction of different functional groups. This was probably because the addition of $-NH_2$, $-NO_2$, and -Br functional groups to the MIL-101(Fe) adsorbent meant that the surface area of the adsorbent was occupied by the functional groups due to their large specific gravity, which reduced the space available for self-use of the adsorbent, in turn resulting in a decrease in the specific surface area and porosity of the adsorbent. Notably, the specific gravity of the $-NH_2$ functional group was smaller than that of the $-NO_2$ and -Br functional groups [38]. Therefore, while the adsorbent with the introduction of -NH2 functional group was slightly better than the adsorbents with the introduction of $-NO_2$ and -Br functional groups, none were as effective as the original MIL-101(Fe).

The adsorption effect on Cr(VI) was observed by adding 10 mL, 20 mL, and 30 mL of formic acid to it while maintaining the optimal preconditions (Figure 6b). The adsorption removal rate increased from 71% to 79% when 10 mL of formic acid was added. The adsorption removal rate of Cr(VI) was reduced to within 10% when the amount of formic acid was increased to 20 mL and 30 mL. The reason for this may be that the structure of the MIL-101(Fe) adsorbent was basically destroyed after the addition of 20 mL and 30 mL of formic acid, and the structural voids collapsed and the porosity was greatly reduced, leading to deterioration of the adsorption effect. The addition of 10 mL of formic acid may

have increased the crystallinity of MIL-101(Fe) while increasing the specific surface area of the adsorbent, which in turn increased the adsorption removal effect of Cr(VI) by less than 10% over the adsorption removal effect of the original MIL-101(Fe) [39].

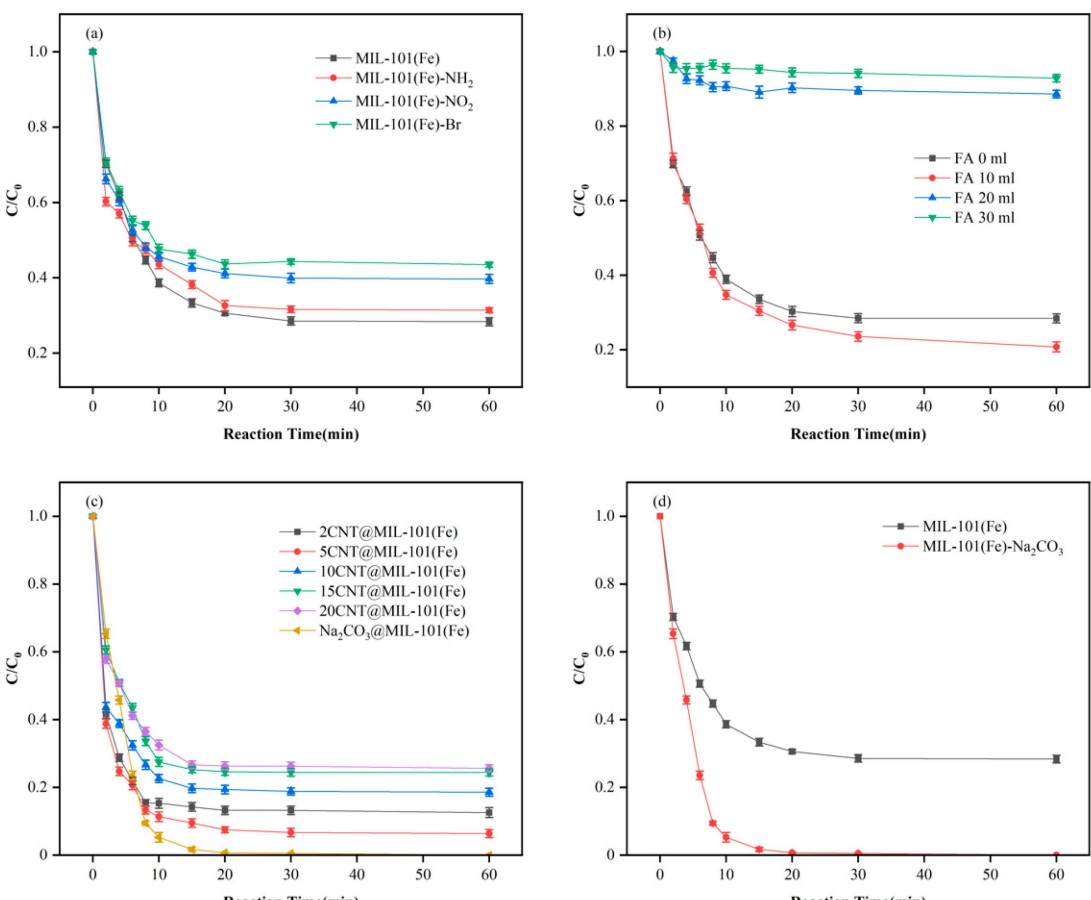

**Figure 6.** (**a**) MIL-101 modification by different groups, (**b**) different FA dosages, (**c**) different CNT dosages, and (**d**) Cr(VI) removal by adsorption with Na$_2$CO$_3$ modification.

Figure 6c shows that the adsorbents with carbon nanotube loadings of 2%, 5%, 10%, 15%, and 20% had adsorption and removal rates of 86%, 92%, 81%, 74%, and 73%, respectively, for the pollutant Cr(VI). The trend of the adsorption and removal rates shows that the best effect was achieved when the loading rate of carbon nanotubes was 5%. The adsorption of carbon nanotubes on the pollutant Cr(VI) is mainly carried out through the functional groups on the surface, and belongs to a potential surface adsorbent, which itself has a certain adsorption effect on Cr(VI). By loading a certain amount of carbon nanotubes on MOFs, it is intended to expand the specific surface area of the adsorbent, thereby increasing the adsorption sites and promoting the adsorption and removal of the pollutant Cr(VI). When the loading rate of carbon nanotubes was 2% and 5%, the adsorption and removal rate of Cr(VI) increased, while when the loading rate was 10%, 15%, and 20% the adsorption and removal rate of Cr(VI) decreased continuously. This may be due to the excessive loading of carbon nanotubes to agglomerate on the surface of MOFs, leading to a poor dispersion effect and covering the adsorption sites on the surface of MOFs, thereby blocking the adsorption pore channel of MOFs and causing the adsorption and removal rate of the adsorbent MOFs for Cr(VI) to decrease [40].

A certain amount of Na$_2$CO$_3$ was added to the adsorbent MIL-101(Fe) as a mineralizing agent. Compared with other mineralizing agents, such as HF and TMAOH substances, Na$_2$CO$_3$ itself is not toxic, and the introduction of Na$_2$CO$_3$ can both improve the crystallinity of the adsorbent and increase the synthetic yield of the adsorbent [41]. It can be

seen from Figure 6d that the adsorption effect of the adsorbent after the addition of $Na_2CO_3$ is significantly better than that of the original adsorbent; the adsorption efficiency reaches 100% under specific conditions, and the adsorption effect has been significantly improved.

It can be seen from Figure 7 that the removal rates of Cr(VI) by adsorption after modification of MIL-101 by different groups take the following order over a time: MIL-101-$Na_2CO_3$ (100%) > 5CNT@MIL-101(Fe) (92%) > MIL-101-FA (79%) > MIL-101 (71%) > MIL-101-$NH_2$ (68%). Therefore, in the modification experiment of the MIL-101(Fe) adsorbent, the best adsorption effect on Cr(VI) was the adsorbent modified by $Na_2CO_3$. In order to further investigate its adsorption characteristics, a single-factor control experiment was carried out with this adsorbent.

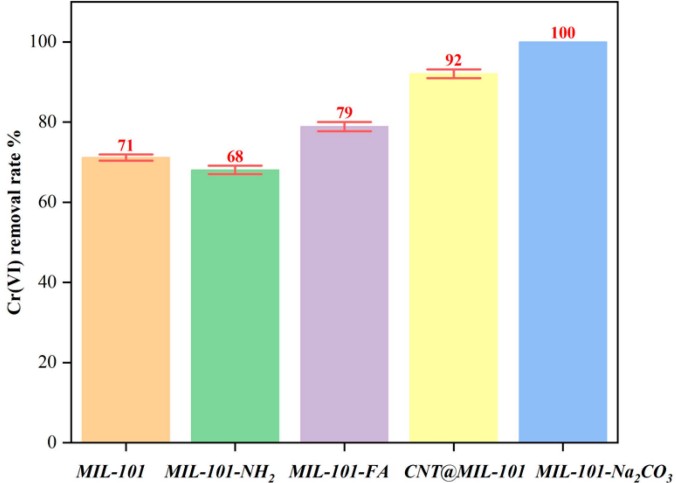

**Figure 7.** Removal rates of Cr(VI) adsorption by MIL-101 modified with different groups.

### 3.3. Adsorption Performance Study

The adsorption experiments were carried out at pH 2, 5, 7, and 9,with an unadjusted solution pH = 5.77 (Figure 8a). The adsorption and removal rates of Cr(VI) pollutants at different pH conditions were 87%, 81%, 80%, 76%, and 74%, respectively, with the highest adsorption and removal rate at pH = 2. The adsorption and removal rates showed a trend of gradual decrease with increasing pH. Chromium mainly exists in aqueous solution in the form of anions such as $HCrO_4^-$, $CrO_4^{2-}$, $Cr_2O_7^{2-}$, etc. [42]. When pH = 2, the content of $H^+$ in water is greater, meaning that a protonation reaction takes place on the surface of the adsorbent and chromium ions are adsorbed to its surface through the effect of electrostatic gravity, which can promote the adsorption and removal of Cr(VI) pollutants. With continuous increase of the pH, more $OH^-$ is generated, which competes with $CrO_4^{2-}$ and other anions for adsorption, causing the adsorption and removal rate of Cr(VI) to decrease [43].

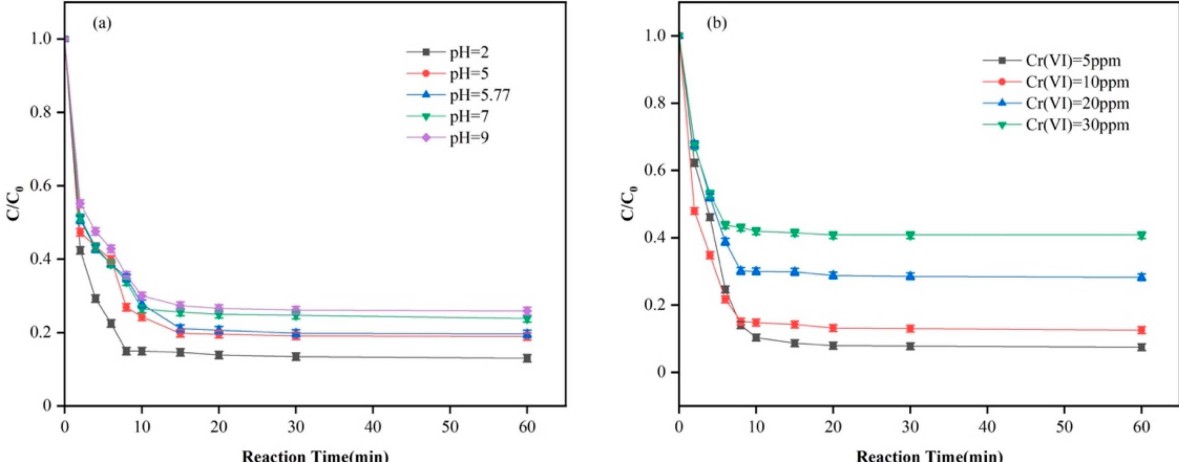

**Figure 8.** (**a**) Effect of different pH levels on the Cr(VI) removal rate and (**b**) effect of different Cr(VI) levels.

In this study, experiments were conducted using Cr(VI) concentrations of 5 mg/L, 10 mg/L, 20 mg/L, and 30 mg/L. The adsorption removal rates shown in Figure 8b indicate that the removal rate decreased from 92% to 59% while the adsorption capacity increased from 4.63 mg/g to 17.42 mg/g with the increasing concentration of pollutants at the same adsorbent dosing.

The adsorption mechanism was explored by adding three anions of different intensities to the adsorption system. It can be seen from Figure 9 that $SO_4^{2-}$ has the greatest effect on the adsorption system, followed by $NO_3^-$, while $Cl^-$ has the least effect. In terms of ionic strength, as the ionic strength of the same ion increases, its inhibitory effect on the adsorption system becomes more obvious, probably because increasing the ionic strength reduces the chance of pollutants coming into contact with the adsorbent, which makes the adsorption effect decrease. Furthermore, because electrostatic force is the key to the adsorption process, the coexisting anions compete for adsorption with the pollutant ions, meaning that less pollutant ions are adsorbed at the active potential on the surface of the adsorbent, in turn leading to a decrease in the pollutant adsorption and removal rates [44].

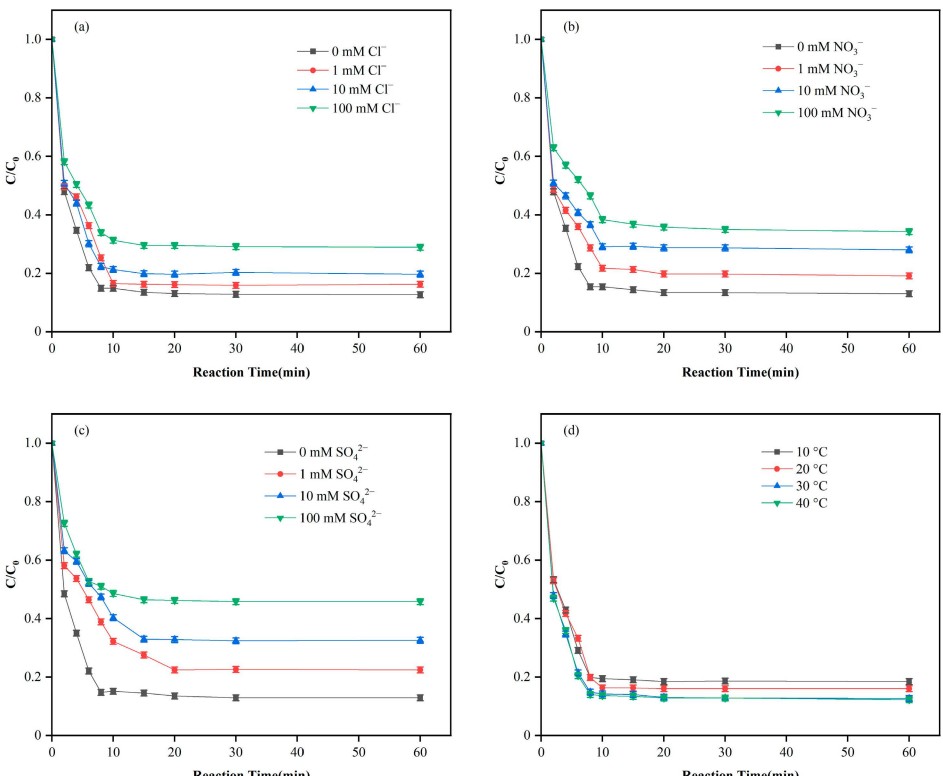

**Figure 9.** (**a**–**c**) Effect of different anion concentrations on Cr(VI) removal and (**d**) effect of temperature on Cr(VI) removal.

From Figure 9d, it can be concluded that the degree of influence of temperature on the adsorption efficiency of the adsorbent is not very large. The adsorption capacity of the adsorbent on Cr(VI) increases slightly with increasing temperature, probably because high temperature increases the frequency of collisions between Cr(VI) and the adsorbent. As the temperature increases, the adsorption removal rate for Cr(VI) increases continuously; from 10 to 30 °C, the adsorption removal rate increases by about 10%, while from 30 to 40 °C the adsorption removal rate essentially does not change. Thus from the economic point of view, a reaction temperature of 30 °C should be selected.

### 3.4. Adsorption Kinetics and Isotherms

Adsorption kinetics is an important factor in the investigation of Cr(VI) adsorption, as it determines the mass transfer rate of adsorption. As shown in Figure 10, based on the adsorption kinetic model (pseudo-second-order and pseudo-first-order), the results of the adsorption experiments were calculated and fitted in order to determine the adsorption mechanism and the control of the adsorption process [45].

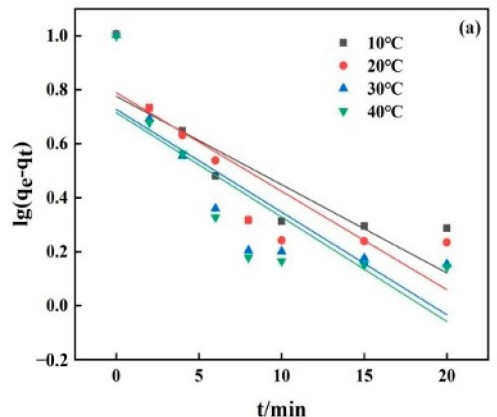 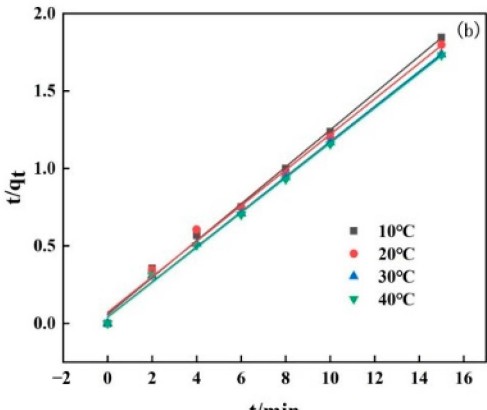

**Figure 10.** (**a**) Pseudo-first-order dynamics model and (**b**) pseudo-second-order dynamics model.

It can be seen from Table 1 that the adsorption process of MIL-101(Fe)-$Na_2CO_3$ on Cr(VI) is more consistent with the proposed secondary kinetic model, with an $R^2$ (0.998) significantly higher than that of the proposed primary kinetic model (0.745). A proposed secondary kinetic model was established at the adsorption rate limiting step, which is mainly used to describe the reactive chemisorption process. This chemisorption process represents the existence of electron sharing or electron exchange between the adsorbent and the pollutant; therefore, it is known from the fitted kinetic model that the adsorption of this adsorbent for Cr(VI) mainly conforms to the proposed secondary kinetics. Analysis of the whole adsorption process of Cr(VI) by the adsorbent can be divided into three stages. The adsorption rate is faster in the beginning stage, slows down gradually as time advances, and finally levels out. The reason for this may be that the concentration of Cr(VI) in the solution is high at first, and the presence of a large number of adsorption sites on the adsorbent surface helps the adsorbent to activate quickly when it comes into contact with the pollutant Cr(VI). This makes the adsorption rate relatively fast; with the gradual extension of the adsorption time, the concentration of Cr(VI) gradually decreases with the combination of adsorption sites, and the adsorption rate of Cr(VI) gradually decreases. The adsorption curve then flattens out, indicating that the adsorption sites on the adsorbent surface are close to saturation.

**Table 1.** MIL-101–$Na_2CO_3$ kinetic model fitting parameters.

| T (°C) | Pseudo-First-Order | | | Pseudo-Second-Order | | |
|---|---|---|---|---|---|---|
| | $K_1$ | $q_e$ | $R^2$ | $K_2$ | $q_2$ | $R^2$ |
| 10 °C | 0.075 | 5.97 | 0.697 | 0.0245 | 8.42 | 0.996 |
| 20 °C | 0.084 | 6.15 | 0.745 | 0.0186 | 8.71 | 0.993 |
| 30 °C | 0.088 | 5.34 | 0.694 | 0.0309 | 8.84 | 0.998 |
| 40 °C | 0.089 | 5.19 | 0.677 | 0.0325 | 8.89 | 0.998 |

Adsorption isotherm models for describing the process of adsorption can be classified into Freundlich, Langmuir, Temkin, and Dubinin–Radushkevich (D-R) models (Figure 11). The Freundlich model assumes that the molecules of the adsorbed substance are adsorbed on the surface of the heterogeneous adsorbent as single or multilayer molecules, and that there is a process of interaction between the adsorbed substance molecules. The Langmuir model assumes that the adsorbent surface is homogeneous, that only one adsorbed substance molecule is adsorbed by the adsorbent on a reaction center with the same energy, and that there is no process of interaction between the molecules of the adsorbed substance. The Temkin model assumes that there is an indirect interaction between the adsorbed substance molecules and the adsorbent. Finally, the D-R model assumes that the adsorbed substance molecules and the adsorbent initially bind by adsorption at the most favorable sites, then subsequently undergo a multilayer adsorption process [46].

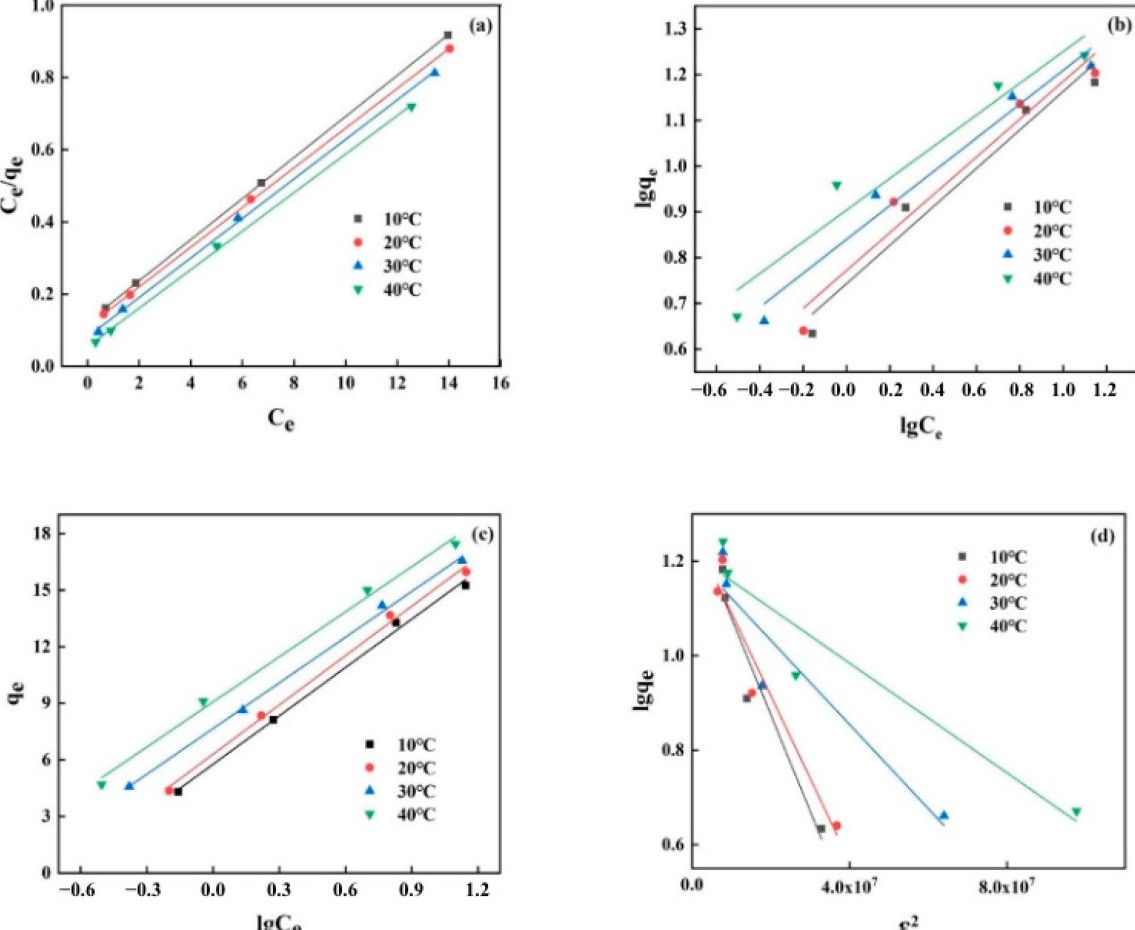

**Figure 11.** (**a**) Freundlich (**b**) Langmuir (**c**) Temkin (**d**) Dubinin-Radushkevich(D-R). Fitting of different isotherm models to the adsorption process.

According to the relevant data, the $R^2$ values obtained from the four models can be fitted separately; it can be seen that the $R^2$ values of the four models range from 0.894 to 0.999, among which the $R^2$ value of the Langmuir adsorption isotherm model is the highest and reaches 0.999, which is the best fit, indicating that the adsorbent conforms to this adsorption model in the process of Cr(VI) adsorption. The $q_m$ (theoretical maximum adsorption amount) of 18.76 mg/g in the Langmuir adsorption isotherm model is close to the adsorption amount of 20 mg/g obtained in the actual experimental process. The simulation results of the remaining three models are slightly worse, with the D-R model the worst, indicating that the adsorbent does not fit well with the other three models during its adsorption of Cr(VI).

The adsorption process of the $Na_2CO_3$–MIL-101(Fe) adsorbent on Cr(VI) is more consistent with the Langmuir adsorption model, indicating that the adsorption mechanism of this adsorbent can be considered as a monolayer adsorption with a uniform surface. Its theoretical maximum adsorption amount is about 18 mg/g, which is similar to the adsorption amount actually sought for this adsorbent. From the data analysis of the Langmuir adsorption separation constant $R_L$, when $R_L > 1$, it is unfavorable for adsorption; when $R_L = 1$, it shows linear adsorption; when $R_L < 1$, it is favorable for adsorption; and when $R_L = 0$, it is irreversible reaction. Therefore, it can be concluded from the above data that $R_L < 1$ favors adsorption.

### 3.5. Thermodynamics and Recycling

The relevant calculations of the thermodynamic parameters are important for the study of spontaneous reactions and thermodynamic changes in the adsorption process. The thermodynamic changes in the adsorption process were understood by calculating the relevant adsorption thermodynamic parameters. When fitting the thermodynamic equations to the adsorption process at four temperatures (10 °C, 20 °C, 30 °C, and 40 °C), the $R^2$ values of the fitted curves reached 0.995 (Figure 12).

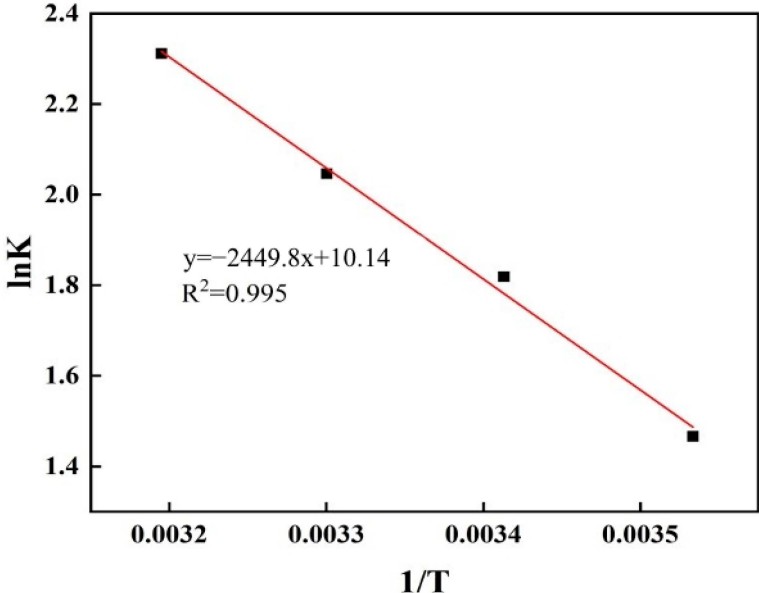

**Figure 12.** Van' Hoff plots of $Na_2CO_3$–MIL-101(Fe) adsorptive Cr(VI).

From the thermodynamic data in Table S4, it can be concluded that the free energy $\Delta G < 0$ indicates that the adsorption process of Cr(VI) adsorption by this adsorbent is feasible and can proceed spontaneously. It can be seen that $\Delta G$ decreases with the increase in temperature, indicating that there is a positive effect on the adsorption process at higher temperatures and that the spontaneity of the reaction is increasing, which is the same as the conclusion obtained from the earlier one-way temperature experiment. Enthalpy $\Delta H > 0$ indicates that the adsorption process of this adsorbent is a heat absorption reaction; entropy $\Delta S > 0$ indicates that the orderliness between the solid–liquid interface present in the adsorption process decreases.

NaOH (0.01 M) solution, HCl (0.01 M) solution, and deionized water were selected as the detergents of Cr(VI); 0.1 g of the adsorbent was added to 50 mL of the above three solutions, then the three mixed solutions were placed in the shaker and shaken for 4 h at room temperature to remove the Cr(VI) pollutants adsorbed on the surface of the adsorbent. The regenerated adsorbent was collected and drained through lyophilizer after 4 h, then the adsorption experiments were performed again. The adsorption removal rate after washing by three kinds of detergent demonstrates that the adsorption effect of regenerated adsorbent on Cr(VI) is NaOH (91.5%) > $H_2O$ (16%) > HCl (2%). These results show that the adsorbent is more stable under acidic and neutral conditions, while desorption occurs best under alkaline conditions (Figure 13).

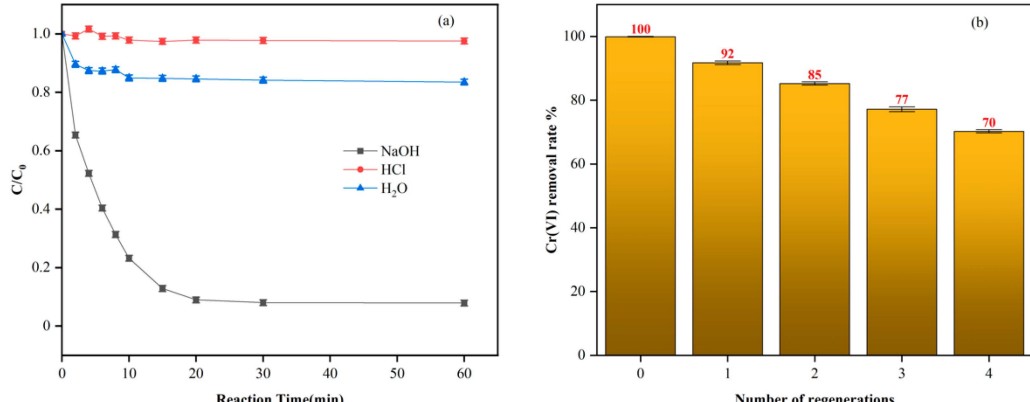

**Figure 13.** (**a**) Selection of eluent and (**b**) cyclic regeneration experiments.

The regenerative availability of MIL-101(Fe)–Na$_2$CO$_3$ was investigated by four repeated adsorption experiments, with 0.01 M NaOH solution selected as the eluent and 0.01 M HCl chosen as the regenerative stabilizer of the adsorbent. After a certain amount of adsorbent had been treated with detergent, the adsorbent was washed three times with HCl to better regenerate the adsorbent, followed by adsorption experiments, and so on, repeatedly. In this way, we conducted four adsorption experiments to observe the adsorption effect of the regenerated adsorbent on Cr(VI). The results showed that the adsorption removal rate of the adsorbent decreased from 100% to 92%, 85%, 77%, and 70% in order, indicating that MIL-101(Fe)–Na$_2$CO$_3$ had better regenerative adsorption capacity for Cr(VI).

Table 2 shows a comparison of the adsorption capacity of several adsorbents for Cr. The data indicate that MIL-101(Fe)-Na$_2$CO$_3$ has relatively good adsorption capacity compared to other adsorbents.

**Table 2.** Comparison of adsorption capacities of MIL-101(Fe)–Na$_2$CO$_3$ for Cr(VI) removal with other recently reported adsorbents.

| Adsorbent | Experimental Conditions | | | Adsorption Capacity (mg/g) | References |
|---|---|---|---|---|---|
| | C0 (mg Cr/L) | Dose (g/L) | Time (h) | | |
| MIL-101(Fe)-Na$_2$CO$_3$ | 10 | 0.5 | 0.3 | 18.76 | This study |
| α-Fe$_2$O$_3$/γ-Al$_2$O$_3$ | 5 | 1.0 | 1 | 3.83 | [47] |
| Nano-γ-Al$_2$O$_3$ adsorbent | 20 | 4.0 | 4 | 13.3 | [48] |
| γ-Al$_2$O$_3$ | 90 | 0.8 | 6 | 6.70 | [49] |
| Activated alumina | 10 | 10.0 | - | 7.44 | [50] |
| Sphere-like γ-Al$_2$O$_3$ | 30 | 1.6 | 4 | 5.70 | [51] |
| Fe-modified T. natans | 20 | 1.5 | 8 | 11.83 | [52] |

In order to judge the selectivity of MIL-101-Na$_2$CO$_3$, we carried out adsorption experiments on other heavy metals. From Figure 14, it can be observed that when adsorbing 10 mg/L of Cr(VI), Pb(II), Cd(II), and V(V), only Cr(VI) was completely adsorbed within 60 min, being adsorbed within 20 min. The adsorption of other heavy metals was the best as well, with Pb(II) reaching 55% within 60 min. For Cd(II) and V(V), the adsorption effect was only the best with Pb(II), reaching 55% within 60 min. Only Pb(II) had the best adsorption effect, reaching 55% within 60 min, while the adsorption effect of Cd(II) and V(V) was only 35% and 28%. The above results indicate that the MIL-101–Na$_2$CO$_3$ adsorbent has good selectivity for Cr(VI).

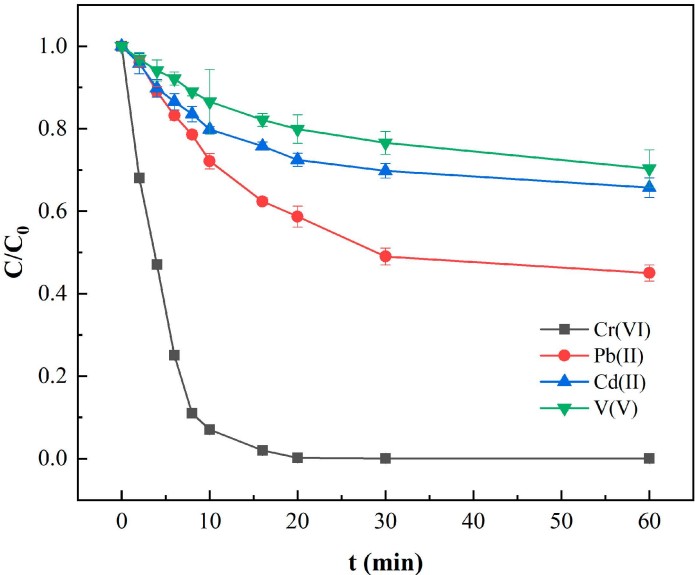

**Figure 14.** Adsorption capacity of MIL-101–NaCO$_3$ on different heavy metals.

*3.6. Mechanistic Analysis*

In order to better verify the MIL-101–Na$_2$CO$_3$ mechanism, the XPS spectra before and after the adsorption reaction using MIL-101–Na$_2$CO$_3$ were obtained, as shown in Figure 15. The full-size XPS spectra of MIL-101–Na$_2$CO$_3$ before the reaction and after adsorption are shown in Figure 15a, where the combination of each element is corrected by the c 1 s peak (284.8 eV). As can be seen in the figure, the general trend of unused and used MIL-101–Na$_2$CO$_3$ in the different systems is consistent, with only minor differences. The two main peaks occupying 711.73 and 724.01 eV are consistent with Fe 2p$_{3/2}$ and Fe 2p$_{1/2}$, as seen in the high-resolution XPS spectrum of Fe 2p, where the satellite peaks of the Fe cation oscillate at 718 eV (Figure 15b). Most of the characteristic peaks shift after the reaction, confirming the involvement of metallic substances in the adsorption reaction.

According to the Na 1 s spectra before and after the catalytic reaction presented in Figure 15c, the presence of the Na 1s peaks verifies the successful participation of Na$_2$CO$_3$ in the MIL-101 system. The changes of the Na 1 s peaks and the shifts of the characteristic peaks before and after the reaction verify that elemental Na plays an important role in the MIL-101-Na$_2$CO$_3$ adsorption system. The O 1 s spectra obtained before and after the reaction show three characteristic peaks corresponding to chemisorbed oxygen, oxygen atom vacancies, and lattice oxygen, respectively. The incorporation of O during the reaction is verified by Figure 15d.

As can be seen in Figure 2f above, the addition of sodium carbonate to MIL-101(Fe) resulted in the growth of a similar crystal-like layer of MIL-101(Fe) from the DMF solution, which grew smaller crystals in the size range of about 200–400 nm.

From Figure S1, it can be seen that the diffraction peaks 2θ = 5.11°, 8.03°, 8.99°, 16.25° for the adsorbent are consistent with the diffraction data described in the literature [29]. The diffraction peaks of the adsorbent before and after adsorption appear at basically the same position, with the peaks broadening slightly after the adsorption reaction, indicating structural defects on the surface of the adsorbent. This might be due to breakage of coordination bonds between the central metal ion and the organic ligand, resulting in reduced surface area of the adsorbent.

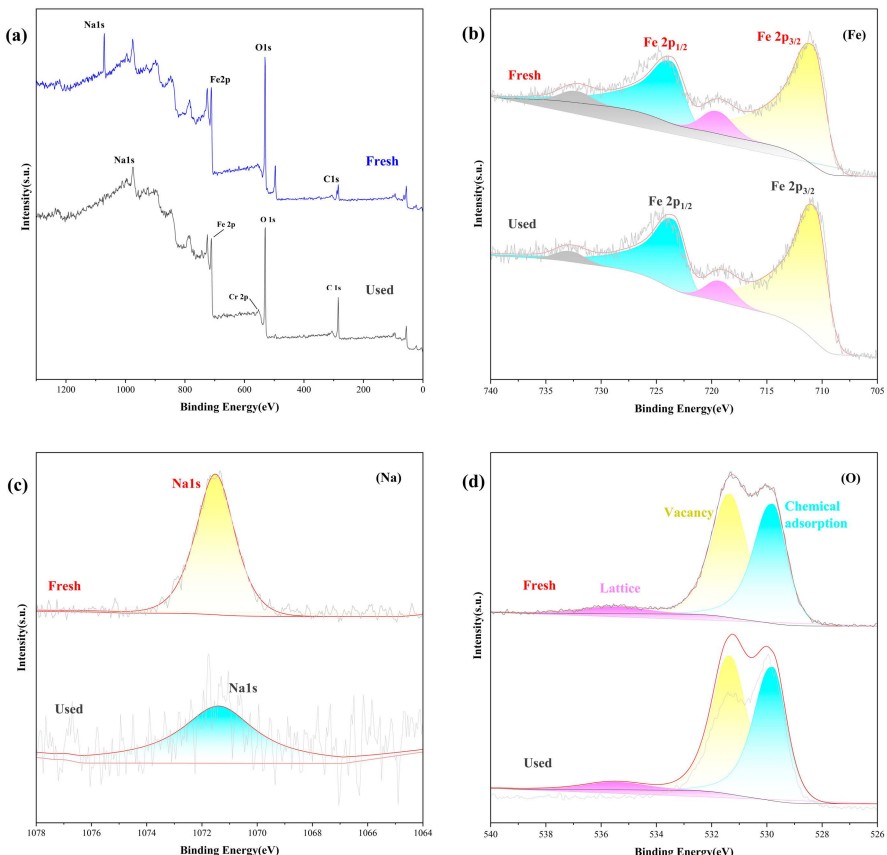

**Figure 15.** High-resolution XPS spectra of (**a**) element spectrum, (**b**) Fe, (**c**) Na, and (**d**) O.

## 4. Conclusions

In this paper, MIL-101(Fe) adsorbent was synthesized for the adsorption of hexavalent chromium pollutants by a hydrothermal method. A series of modifications were carried out in order to observe the induced adsorption effects: MIL-101-$Na_2CO_3$ > 5CNT@MIL-101(Fe) > MIL-101-FA > MIL-101 > MIL-101-$NH_2$. In the adsorption process of MIL-101–$Na_2CO_3$, the adsorption equilibrium time for hexavalent chromium was 20 min, the optimum pH for the adsorption experiment was 2, and the adsorption process was mainly electrostatic adsorption. The optimum temperature of the adsorption reaction was 30 °C, and the adsorption removal rate showed an increasing trend during the temperature change from 10 °C to 40 °C. These results indicate that the adsorption experiment involved a heat absorption reaction. From the fitting results of the kinetic model and isotherm model, the adsorption process is consistent with the Langmuir model of monomolecular layer adsorption and the proposed secondary kinetic model of chemical bonding force. Based on the characterization of the adsorbent material, the adsorbent showed structural residues and a decrease in specific surface area after the reaction, which is consistent with a decrease in adsorption capacity.

The successfully modified MIL-101(Fe)–$Na_2CO_3$ material has potential for development in the treatment of Cr(VI) wastewater. Although preliminary progress has been made in the treatment of Cr(V) in wastewater in this paper, there remain many issues to be further explored, including the following:

1. While the modified MIL-101(Fe)–$Na_2CO_3$ material has good prospects, its stability needs to be improved, and the preparation method needs to be improved in future work to find a simpler and more economical experimental preparation method.
2. Future work could explore different alloy materials to replace the single metal, reducing the cost of preparing the f materials while providing improved adsorption performance.

3. Other carrier materials could be explored in order to find the most effective carrier and further optimize the removal of Cr(VI).
4. The significant advantages of metal–organic frameworks can be utilized to broaden the modification methods and improve the adsorption capacity for a variety of heavy metals.

**Supplementary Materials:** The following supporting information can be downloaded at: https://www.mdpi.com/article/10.3390/w16010025/s1, Figure S1: XRD pattern of MIL-101-Na$_2$CO$_3$; Table S1: BET results of MIL-101(Fe), MIL-101(Fe)-NH$_2$, MIL-101(Fe)-NO$_2$, MIL-101(Fe)-Br, CNT@MIL-101(Fe); Table S2: Effect of the ratio of metal to organic ligand synthesis on the removal rate of Cr(VI); Table S3: Effect of different DMF dosage on the removal rate of Cr(VI); Table S4: MIL-101-Na$_2$CO$_3$ thermodynamic model fitting parameters.

**Author Contributions:** Conceptualization, C.W. and J.C.; validation, C.W., J.C. and Q.Y.; formal analysis, C.W.; data curation, C.W. and J.C.; writing—original draft preparation, C.W.; writing—review and editing, Q.Y. All authors have read and agreed to the published version of the manuscript.

**Funding:** This research received no external funding.

**Data Availability Statement:** The data presented in this study are available on request from the corresponding author.

**Conflicts of Interest:** The authors declare that they have no known competing financial interest or personal relationships that could have appeared to influence the work reported in this paper.

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
