# Peer review of "Rapid Removal of Cr(VI) from Wastewater by Surface Ionized Iron-Based MOF: Ion Branching and Domain-Limiting Effects"

_water, doi:10.3390/w16010025_

Round 1
Reviewer 1 Report
Comments and Suggestions for Authors
The following minor corrections are needed:
Result & Discussion
3.1 Characterization
Line 148 In addition , the sioze of the CNT@MIL-10(Fe) increased compared to
Line 153 nanotubes were almost the same as
Line 155 did not prevent the generation of MIL-10 crystals
Line 164 peaks were also present in the remaing four materials
Line 436 Figure
Line 442 Figure
Comments on the Quality of English LanguageThe quality of english language was adequate with minor corrections as indicated above.
Reviewer 2 Report
Comments and Suggestions for Authors
Dear Authors,
Kindly consider the following comments and concerns:
- The abstract does not need to delve into the three stages of adsorption. Ensure it aligns with the criteria of an abstract.
- Clearly articulate the novelty of the article.
- Specify the sources of the chemicals and provide their specifications.
- In Figure 2, I couldn't locate slide e at 200 nm. Maintain consistency in presenting the results.
- Compare your results with other products in the market/literature for context.
- Clarify the control used in this research.
- Provide information on the recommended capacity of the adsorbent for commercial use.
- Outline recommendations for future research.
- No investigate of the sensitivity and selectivity of the product has been reported.
- If possible, provide SEM for the materials before and after usage.
I recommend that the authors review the article and make minor edits.
Reviewer 3 Report
Comments and Suggestions for Authors
In this study, adsorbents consisting of a metal and a ligand for chromium are prepared, their adsorption capacity is evaluated and steric conformation inhibition is discussed. The content of the paper is based on an academic description and is basically unproblematic. It is stated that hexavalent chromium is removed or adsorbed, but it should be explained whether it is reduced to trivalent chromium and then adsorbed, as discussed in many reports, or whether the total chromium remains the same but appears to be reduced by the reduction.
Equations (1) and (2) are not aligned.
The 'sodium number of sodium carbonate (Na2CO3)' in L432 is not under subscripted.
Round 2
Reviewer 2 Report
Comments and Suggestions for Authors
The authors have made diligent efforts to address the comments provided by the reviewers.
Comments on the Quality of English LanguageThe English language is acceptable.